# Evaluation of Nano-Mechanical Behavior on Flax Fiber Metal Laminates Using an Atomic Force Microscope

**DOI:** 10.3390/ma12203363

**Published:** 2019-10-15

**Authors:** Zehua Qu, Xiaoxia Pan, Xiaoyue Hu, Yichun Guo, Yiou Shen

**Affiliations:** 1State Key Laboratory of Molecular Engineering of Polymers, Department of Macromolecular Science, Fudan University, 2205 Songhu Road, Shanghai 200433, China; quzehua@fudan.edu.cn (Z.Q.); panxiaoxia@fudan.edu.cn (X.P.); huxy@fudan.edu.cn (X.H.); 2School of Aerospace Engineering and Applied Mechanics, Tongji University, 1239 Siping Road, Shanghai 200092, China

**Keywords:** plant fiber, fiber metal laminates, surface treatments, nano-mechanical behavior, atomic force microscopy

## Abstract

The application of plant fiber-reinforced composite (PFRC) is limited due to its relatively low mechanical properties. The hybridization of a thin metal layer with plant fiber into a fiber metal laminate can largely improve the mechanical performance and the brittle fracture behavior of PFRC. However, both plant fiber and metal have difficulty bonding with the polymer matrix. In this paper, several different surface treatment methods were applied on Al alloy sheets, and the influence of surface treatments on the surface morphology and nano-mechanical properties of Al alloy were studied using an atomic force microscope (AFM). After the preparation of flax fiber–metal laminates (FFMLs) with a vacuum-assisted resin transfer molding (VARTM) technique, the nanomechanical properties of different modified FFMLs were also evaluated with an AFM. It was found that the surface treatment combination of the sulfuric acid-ferric sulfate-based treatment (P2 etching) and the silane coupling agent provided the best adhesion force and modulus for Al alloy sheets at nanoscale resolution, which contributed to the surface energy increasing and strong covalent bonds between metal and polymer matrix. The resulting manufactured FFMLs also exhibited the highest nano-mechanical properties due to the great improvement of interfacial properties between metal and matrix, which was caused by mechanical interlocking mechanism and covalent bonds between metal/fiber and resin. Macromechanical performance, including tensile and flexural properties of these modified FFMLs, was also investigated. Comparison of the modulus at the nanoscale and macroscale showed reasonable agreement, and it revealed the tough interlaminar mechanisms of these types of FFMLs.

## 1. Introduction

In recent years plant fiber has been utilized as an environmentally friendly reinforcement to replace asbestos and fiberglass in strengthening thermosetting and thermoplastic polymers for various applications [1,2]. However, compared to conventional composites, such as glass fiber reinforced plastic (GFRP), the main problem with plant fiber reinforced plastic (PFRP) is the relatively low mechanical properties [3]. Therefore, the application of plant fiber composites is still limited to non-structural components, e.g., panels, ceilings, partition boards, and interior car components. To promote the development of the application of PFRPs, the improvement of the mechanical properties is urgent, and further knowledge about the strengthening mechanisms is required.

Hybridization is one of the most effective methods to improve the strength, modulus, and other mechanical properties of the composites. It has been proved that the hybridization of the same fiber volume fraction of high-performance fiber with plant fiber can significantly improve the tensile modulus of PFRPs by more than 40% [4]. However, the low impact damage tolerance and brittle failure feature of the hybrid composite are still not changed. In recent decades, fiber–metal laminates (FMLs) made by thin metal alloy sheets and fiber-reinforced composite piles have been wildly used in the aerospace field. This composite integrates the characteristics of both fiber composite and metal material and exhibits excellent fatigue performance, damage tolerance, and high impact strength [5]. In the past few years, some attempts have been made on hybridizing natural fiber with metal sheets for natural fiber metal laminates, and the fibers include basalt, jute, sisal, and so on [6,7,8,9]. It was found that this method can greatly improve the mechanical properties of the composite. However, the mechanical behavior of FMLs is not only related to the properties of the fiber, metal, and matrix components themselves but is also greatly affected by the interfacial bonding between fiber/matrix and metal/matrix. Previous work [10] has found PFRP exhibits inferior interfacial properties due to the weak bonding between the hydrophobic resin and the hydrophilic plant fiber. Therefore, it is common to apply a surface treatment on plant fibers before using them as reinforcement in composites, which can enhance the bonding between fiber and matrix. Therefore, the stress transfer in the composite will also be improved [11,12]. It was found that the silane coupling agent treatment can effectively improve the interface adhesion and mechanical properties of plant fiber-reinforced polymer composites through the formation of covalent bonds between fiber and resin [13]. Surface treatment of metals is also one of the most significant key factors affecting the properties of metal/matrix bonds [14]. A variety of surface treatment methods have been introduced for enhancing the interface adhesion between metals and polymer matrix, including mechanical, chemical, electrochemical, coupling agents, etc. [15,16]. The use of surface treatment methods on fibers or metal, before laminate manufacturing, can significantly modify the surface morphology, surface energy, and wettability of the metals and, consequently, the mechanical properties of FMLs. Some of the modifications on the metal surface are in microscale, even nanoscale. Therefore, corresponding to these modifications, microscale or nanoscale characterization methods have to be used to investigate the interface adhesion properties and reveal the mechanisms of FMLs. However, there are few studies on the nanoscale characterization and nanomechanical properties of FMLs.

With the development of atomic force microscopy, several researchers have applied the atomic force microscope (AFM) technique to access mechanical properties of glass fiber reinforced composite laminates at submicron or nanoscale [17,18,19]. These works revealed a homogeneous behavior of glass fibers, and the modulus obtained through these methods is consistent with the modulus obtained by tensile tests. Bourmaud et al. [20] and Chegdani [21] used AFM techniques to measure the modulus of plant fibers. But they found the elastic modulus obtained by AFM was not consistent with the values obtained by tensile tests, which might be due to the heterogeneity of plant fibers. The mechanisms of this phenomenon are still unclear. However, these studies implied the complex relationship between material structure and mechanics at the nanoscale. Therefore, it is essential to evaluate the nanomechanical properties of composite materials and to find a connection with their macromechanical properties.

In this paper, several surface modifications were undertaken on both flax fiber and aluminum alloy sheets before the manufacturing process to improve the interlaminar properties of the flax fiber–metal laminates (FFMLs) composites. The surface morphology of the Al alloy sheets and the nanomechanical of modified FFMLs were investigated using an atomic force microscope, and the surface treatment influence mechanisms on the interlaminar properties of this FFMLs composite were also revealed.

## 2. Materials and Methods

### 2.1. Materials

The unidirectional flax fabric was provided by LINEO Co. Ltd., Meulebeke, Belgium (the density was 1.5 g/cm^3^, the thickness was 0.18 mm, the areal density was 200 g/m^2^), the aluminum alloy sheet Al2024-T3 was provided by Nippon Light Metal Co., Ltd., Tokyo, Japan (the density was 2.7 g/cm^3^, and the thickness was 0.2 mm). The epoxy resin (NPEL-128) system was provided by Beijing Colas Chemical Technology Co., Ltd. (Beijing, China). It was mixed with curing agent and accelerating agent at the weight ratio of 100:80:1.

### 2.2. Surface Treatments

A 2% silane coupling agent aqueous solution was applied on the surface pre-treatment of flax fiber fabric. The fabric was soaked in the solution for 20 min. After that, the moisture was removed in an oven at 100 °C for 4 h, leaving strong covalent bonds between fiber and the silane coupling agent [13].

Several surface pretreatment methods were employed to modify the aluminum alloy sheets, including mechanical abrasion, alkaline treatment, sulfuric acid-ferric sulfate-based treatment (known as P2 etching), and silane coupling agent treatment. Holes were drilled into all the Al alloy sheets to produce flow paths for the resin to permeate through the metal layer during the manufacturing process. The holes were carefully drilled with a uniform diameter of 1 mm, and the distribution of holes is shown in Figure 1. Then the metal sheet surfaces were cleaned with acetone to degrease the surface before any surface pretreatments were carried out. The following surface pretreatments were employed to modify the surfaces of the aluminum alloy:

T1. Mechanical abrasion: The Al alloy sheets were ground and polished along the rolling direction of metal, and were cleaned by anhydrous ethanol at room temperature.

T2. Alkaline treatment: The Al alloy sheets were soaked in an alkaline solution bath at 60 °C for 1 min (30 g/L sodium hydroxide and 30 g/L sodium carbonate), and were washed with clean water and then dried [22].

T3. P2 etching: The Al alloy sheets were immersed in a water solution with 185 mL/l sulfuric acid (97%) and 127 g/L ferric sulfate, at 65 °C for 8 min, and rinsed in tap water (percentages by weight: 48% H_2_O, 37% H_2_SO_4_ and 15% FeSO_4_) [15].

T4. Silane coupling agent treatment: The Al alloy sheets were soaked in 2% 3-Phenyl-aminopropyl-trimethoxy-silane aqueous coupling agent solution for 10 min, and finally dried in an oven at 100 °C for 1 h [23].

Table 1 gives the surface treatment combinations that were applied in this study.

### 2.3. Laminate Fabrication

Silane coupling agent-treated unidirectional flax fiber fabric and Al alloy treated with five different methods, as listed in Table 1, were employed in fabricating FFMLs using a standard vacuum-assisted resin transfer molding (VARTM) technique [10]. The layups of flax fabrics and thin Al alloy sheets were placed on the top of a smooth glass table between two sheets of peel plies. The resin system was degassed in a vacuum oven for 30 min to remove air bubbles before perfusion. Once the vacuum was surely set, the resin was impregnated in the mold, and the flow was evenly distributed across the plate. The laminates were then cured at room temperature for 24 h and post-cured at 120 °C for 2 h. The stacking sequence of all the manufactured FFML laminates was [Al/0°/90°/Al/90°/0°/Al] and the fiber volume fraction and metal volume fraction were 22% and 18%, respectively. Flax fiber-reinforced plastic (FFRP) laminates with a stacking sequence of [0°/90°]_2s_ were also manufactured for comparison. The parameters of the resulting laminates are given in Table 2.

### 2.4. Material Characterization

The surface morphology and microstructure of flax fiber and metal were examined using a field emission scanning electron microscope (Zeiss, Ultra55, Kohen, Germany) and an atomic force microscope (FastScan Microscope, Bruker, Beerlika, MA, USA).

The nanomechanical properties of the metal sheets and composites were mapped by the mode of PeakForce quantitative nanomechanical mapping (PF-QNM) of the AFM [24,25]. A Bruker tip named RTESPA-300 with a cantilever length of 125 mm, radius of 8 nm, and resonance frequency of 300 kHz was used. The deflection sensitivity, the exact spring constant of the cantilever, and the tip radius were calibrated by the standard sample. The metal sheet was cut into 5 × 5 mm^2^ for examination. The FFMLs composites were cut into 5 × 10 mm^2^ and cast into a mold with epoxy. The cross-section of the cast samples was carefully ground for further examination.

Tensile and flexural tests were performed on both FFML and FFRP samples using a universal testing machine (CSS-20N, Shenzhen Wance Co. Ltd., Shenzhen, China), according to ASTM D638 and ASTM D790, respectively.

## 3. Results

### 3.1. Surface Morphology of Materials

First, the silane treatment on flax fiber caused the formation of covalent bonds between the organic functional group of the silane and the hydroxyl of flax fiber. A silane film was found formed on the surface of the flax fiber, as shown in Figure 2. The interfacial adhesion between untreated flax fibers and hydrophobic epoxy was very weak due to the presence of hydroxyl and other polar groups in the flax fiber. The hydrophilic nature of the flax fibers can be minimized with silane coupling agent treatment. It can increase the wettability of the fibers within the matrix and increase the interfacial bond strength. There have been many studies that prove that silane treated plant fiber and epoxy can form interpenetrating polymer networks using XPS and AFM characterization methods, and the interface adhesion between flax fiber and resin within the composite are clearly enhanced [26,27].

Different surface treatment methods have different mechanisms for improving the bonding capability between metal alloy and polymers. The surface morphology of Al2024-T3 sheets after five combinations of surface treatments are presented with 3D AFM height images in Figure 3. The AFM height image of the untreated Al sheet is also given for comparison. The depth of the color represents different heights, dark color indicates a relatively low height, while light color indicates higher height. Mechanical polishing (MP) can increase the surface roughness of Al alloy sheets; therefore, the surface energy is enhanced, and this would provide mechanical interlocking between metal and epoxy, thus increasing bond strength. It can be observed from the 3D AFM height map, Figure 3A1, that the initial surface of untreated Al alloy was non-uniform, which causes the high roughness value in Table 3. Actually, this non-uniform surface goes against the bonding between metal and epoxy. Abrasive grinding produced more uniform grooves on the Al alloy surface, as shown in Figure 3B1. Alkali and P2 etching created a porous structure on the Al surface, which also provided mechanical interlocking by increasing the surface roughness, as shown in Figure 3C1,E1. The thin oxide layer on Al alloy was soluble, and the Al alloy was corroded by strong acids and alkalis, which left microscale or nanoscale pits uniformly distributed on the Al surface, as shown in Figure 4. Prolongo et al. [15] found that Al2024 alloy was easily affected by alkaline treatment due to the presence of intermetallic compounds with different electrochemical potential, which was preferentially attached with alkali. They also found that P2 etching treatment caused a slightly decrease in the density of Al2024 and the variation of the elemental composition of the treated Al alloy. Silane treatments were applied after alkali and P2 etching. It can be observed in Figure 3D1,F1 that the roughness of the silane treated samples reduced compared to that of mechanical polishing and alkali solution immersion (MPAL) and P2, indicating that a silane film was formed on the Al surface. Hamill and Nutt [28] used XPS to confirm the presence of a silane film on the Al2024 surface after silane treatment. The roughness data, Rq and Ra, of various surface-treated Al sheets obtained using AFM is shown in Table 3.

### 3.2. The Effect of Surface Treatments on Nanomechanical Properties of Al Alloy

Young’s modulus of Al alloy with different surface treatments can also be obtained using AFM [24,29]. The Derjaguin–Muller–Toporov (DMT) model accommodates adhesion forces during contact, and is suitable for testing stiff materials with low adhesion [30]. The DMT modulus and the modulus distribution graphs of Al alloy are presented in Figure 5. Figure 5A2–F2 show the modulus image of Al alloy sheets with different surface treatments, again the depth of color represents the variation of modulus. It can be found from these images that the modulus at different locations on the Al alloy surface was varied, and the distribution modulus graphs corresponding to the modulus maps are given in Figure 5a2–f2.

It can be observed that the modulus values of the untreated Al alloy were mainly distributed on two points (1.5 MPa, 2.5 MPa) and show a shoulder peak in the distribution curve, as shown in Figure 5a2. However, the modulus distribution graphs of all the modified Al alloy show a single peak. The P2Si treated Al alloy processed the highest modulus and distribution range among untreated and modified Al alloy sheets. These results suggest that the modulus of Al alloy can be improved with P2Si surface treatment.

The AFM adhesion images in Figure 6A3–F3 reflect the level of adherence force on the untreated and other five surface modified samples. Figure 6a3–f3 show the adhesion force distribution graphs of these six types of Al alloys. It can be observed from Figure 6a3 that the adhesion force of the untreated Al alloy was approximately 100 nN, the adhesion force values of all six type of Al alloys can be arranged as P2 > P2Si > Al > MPAL > MP > MPALSi. This result indicates that P2 etching treatment on Al alloy provided the best adhesion capacity for the Al alloy surface due to the microscale and nanoscale pits caused by acid etching.

### 3.3. Effect of Surface Treatments on Nanomechanical Properties of FFMLs

The cross-section morphologies of the FFRP and five types of modified FFMLs were evaluated with 2D AFM height images, which revealed the interlaminar adhesion status of different laminates. The AFM height images of the FFRP and five types of modified FFMLs are shown in Figure 7. The white dash lines on graphs represent the interface between the metal layer (the left side) and the FFRP layer (the right side). The existence of cavities between metal and FFRP layers in MP-FFMLs and MPAL-FFMLs was quite obvious, as shown in Figure 7B1,C1. Some smaller cavities were observed in MPALSi-FFMLs and P2-FFMLs, as shown in Figure 7D1,E1. There was no apparent space between the metal layer and FFRP layer for the P2Si-FFMLs sample, as shown in Figure 7F1, this suggesting the interlaminar interfacial properties of FFMLs has been significantly improved by this surface treatment method.

The AFM DMT modulus mapping images of the FFRP and the five types of modified FFMLs are given in Figure 8A2,F2, and the modulus distribution graphs of these laminates are given in Figure 8a2–f2. It can be observed from Figure 8g2 that the modulus of FFRP was below 2.5 GPa and mainly distributed at approximately 0.8 GPa. However, it can be observed from Figure 8b2–f2 that the induction of Al alloy greatly changed the modulus and the distribution range of the laminates. The modulus of FFMLs with MP, MPAL, and MPALSi surface treatments was lower than that of the FFRP due to the weak bonding between metal and FFRP layers. The modulus of P2-FFMLs and P2Si-FFMLs composite was improved compared to that of the FFRP, and P2Si-FFMLs exhibited the highest value, mainly distributed on approximately 1.8 GPa, which was a 125% enhancement compared to that of the FFRP laminate, and the modulus distribution range was also very wide, which achieved more than 10 GPa. These results indicate that the hybridization of proper surface-treated Al alloy sheets with flax fiber fabrics can significantly enhance the modulus of composites at the nanoscale.

The AFM adhesion images and the adhesion force distribution graphs of FFRP and five types of modified FFMLs are shown in Figure 9A3–F3,a3–f3, respectively. The adhesion force of FFMLs was much higher compared to the modified Al alloy. It can be found from Figure 9a3 that the adhesion force distribution of FFRP was mainly on 0.24 μN. Inducing the various surface-treated Al alloy sheets greatly enlarged the adhesion force distribution range. It is clear that the adhesion force of P2Si-FFML was concentrated on approximately 1.3 μN, which increased 490% compared to that of the FFRP composite, and the trend of adhesion force results for the six types of the composite is consistent with that of the DMT modulus.

### 3.4. Effect of Surface Treatments on Macromechanical Properties of FFMLs

It can be observed from Figure 10 that the tensile strength and modulus of all FFMLs were higher than those of FFRP due to the inducing of the Al alloy sheets. However, the improvement of tensile strength was not significant, which varied from 23% to 31%. Remarkably enhancements on modulus were observed for all FFMLs compared to that of the FFRP, which varied from 134% to 357%. The P2Si-FFMLs showed the highest modulus and relatively high strength due to the improvement in interlaminar interface performance. The fracture morphologies of different samples after tensile failure are shown in Figure 11. It can be observed that the fracture section of the FFRP was relatively flat, and the fiber fracture was the damage mode. Seriously delamination was observed during tensile tests on MP-FFMLs, MPAL-FFMLs, and MPALSi-FFMLs, and successive fractures occurred for different layers within the laminates. However, as for the P2-FFMLs and P2Si-FFMLs, in spite of some delamination that emerged when the samples were being tensioned, all the layers were fractured in the center of the sample, and the fracture section was flat due to the improvement of the interlaminar properties. In addition, the tensile modulus of FFRP and five types of modified FFMLs showed good agreement with the DMT modulus of these composites, for which the P2Si-FFMLs provided the highest modulus among these composites.

Three-point bending tests were conducted on these laminates, and the effect of different surface treatments on flexural strength and modulus of FFMLs are shown in Figure 12. The FFMLs provided higher flexural strength and modulus compared to that of the FFRP as expected, and the effect of MP, MPAL, and MPALSi surface treatment methods on improving the flexural properties of FFMLs was not significant. However, it can be observed from Figure 12 that the flexural strength and modulus of P2-FFMLs and P2Si-FFMLs were greatly enhanced, and the highest strength and modulus were 56% and 40% higher than that of the FFRP, respectively. The side view of FFRP and FFMLs following three-point bending tests are given in Figure 13. It can be observed that the non-P2 etching treatment groups showed serious delamination. And it is clear that the P2-FFMLs and P2Si-FFMLs show better interlaminar interface properties. Delamination was not obvious for these two laminates following the flexural test.

## 4. Discussion

The preliminary analysis shows that the addition of a high modulus aluminum plate can effectively improve the tensile elastic modulus of flax fiber laminates. However, due to poor interfacial properties, the failure mode of some experimental groups was not fiber or metal fracture but delamination. It can be found from the results that the combination of P2 etching and silane coupling agent treatment gave full play to the advantages of a high modulus of the metal because of the increased bonding power between layers. This is because P2 etching presented a high porosity surface for the Al alloy [15]. Hamill and Nutt [28] found that the silane treatment did not substantially increase wettability or roughness of the Al alloy surface. However, it can improve the bonding ability between metal and epoxy through forming a covalent bond. Therefore, the P2Si treatment can improve the interlaminar bonding of the Al alloy layer with epoxy more effectively and make the FFMLs respond to the tensile force more like an integrity, avoiding reducing the bending performance due to the existence of the layering in the bending stress.

AFM data further indicated that MP treatment can remove the weakly adhered layer and the contaminated parts on the surface of the aluminum, exposing the new oxidized uniform adhesive surfaces, providing a lower surface roughness compared with the untreated surface of the aluminum alloy. While after the alkali treatment, the roughness increased due to the removal of the alumina film from the surface of the aluminum [15]. The silane coupling agent can make the roughness decrease compared with that of the coupling agent untreated surfaces. Furthermore, the same situation of the P2 etched surfaces was found: The P2-treated aluminum alloy had a higher degree of roughness while P2Si showed a slightly lower roughness compared with P2-treated aluminum alloy. Meanwhile, the adhesion forces of the silane coupling agent treated surfaces were also found to be decreased compared with that of the untreated surfaces. Therefore, it is concluded that the combination of porous structure and silane film on the Al alloy surface contributes to the improvement of the modulus of Al alloy and the interlaminar properties of FFMLs rather than the adhesion ability and the roughness. Further, with this understanding, it was found that the good binding performance of the composites can be directly investigated in the height images of the AFM. The adhesion force of the silane coupling agent treated FFMLs can be improved when modified aluminum alloy was formed into FFMLs, and the P2 etching modified FFML showed better mechanical properties, which were shown from the modulus and the adhesion data. The inconspicuous dividing lines between aluminum and FFRP on the surfaces of P2-FFML and P2Si-FFML in Figure 7E1,F1 indicated the high binding strength. The adhesion force data in Figure 9 also indicates that P2Si-FFMLs have the best interlaminar properties.

## 5. Conclusions

In this paper, several different surface treatment methods were applied on Al alloy sheets, and the influence of surface treatments on the surface morphology and nanomechanical properties of the Al alloy were studied using an atomic force microscope in PeakForce QNM mode. It was found that P2Si surface modification provided the best adhesion force and the nanoscale modulus for Al alloy and its composite laminate than that of the other surface modification. The results of the AFM showed that this method improved the interlaminar properties of FFMLs by increasing the surface energy by creating a porous structure and forming covalent bonds on the surface of Al alloy. Moreover, the macromechanical properties of FFMLs showed reasonable agreement with the nanomechanical properties. It can be concluded that AFM is a useful characterization tool for the evaluation of mechanical and interfacial properties of laminate composites.

## Figures and Tables

**Figure 1 materials-12-03363-f001:**
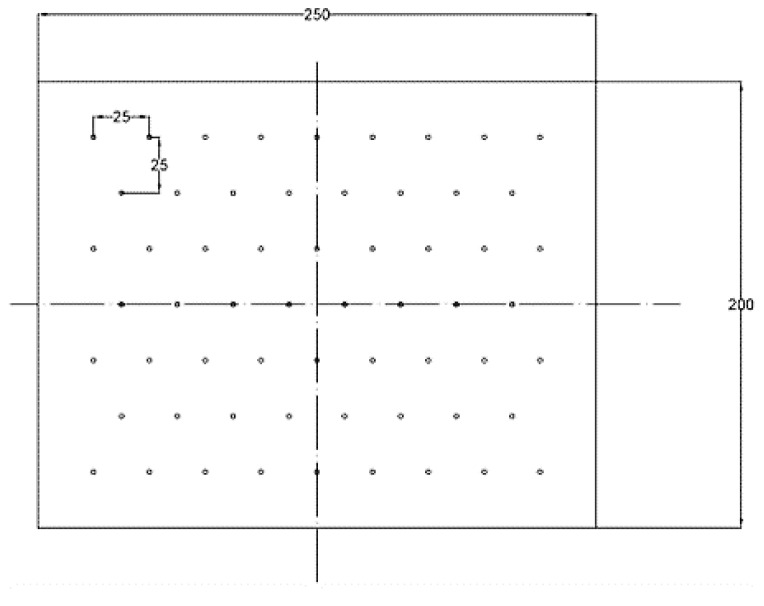
Hole-map design of the Al alloy sheet.

**Figure 2 materials-12-03363-f002:**
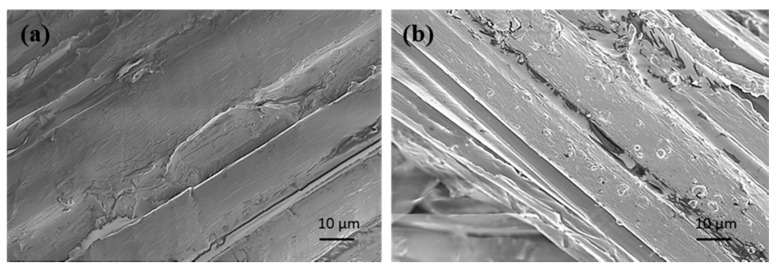
The surface morphology of flax fiber (**a**) untreated and (**b**) silane treated.

**Figure 3 materials-12-03363-f003:**
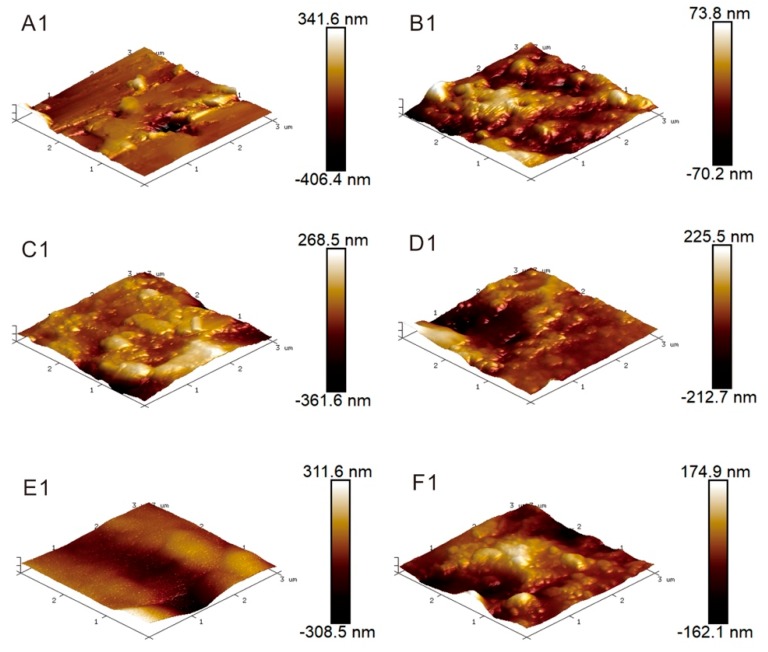
3D atomic force microscope (AFM) height images of the Al alloy with different surface treatment, (**A1**) untreated, (**B1**) mechanical polishing (MP) treated, (**C1**) Mechanical polishing and the alkali solution immersion (MPAL) treated, (**D1**) Mechanical polishing, alkali solution immersion, and silane coupling agent surface (MPALSi) treated, (**E1**) P2 etching (P2) treated and (**F1**) P2 etching and silane coupling agent surface (P2Si) treated Al alloy.

**Figure 4 materials-12-03363-f004:**
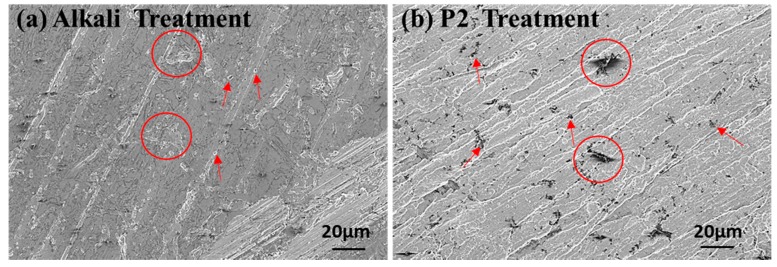
Surface morphology of (**a**) alkali-treated and (**b**) P2 treated Al alloy.

**Figure 5 materials-12-03363-f005:**
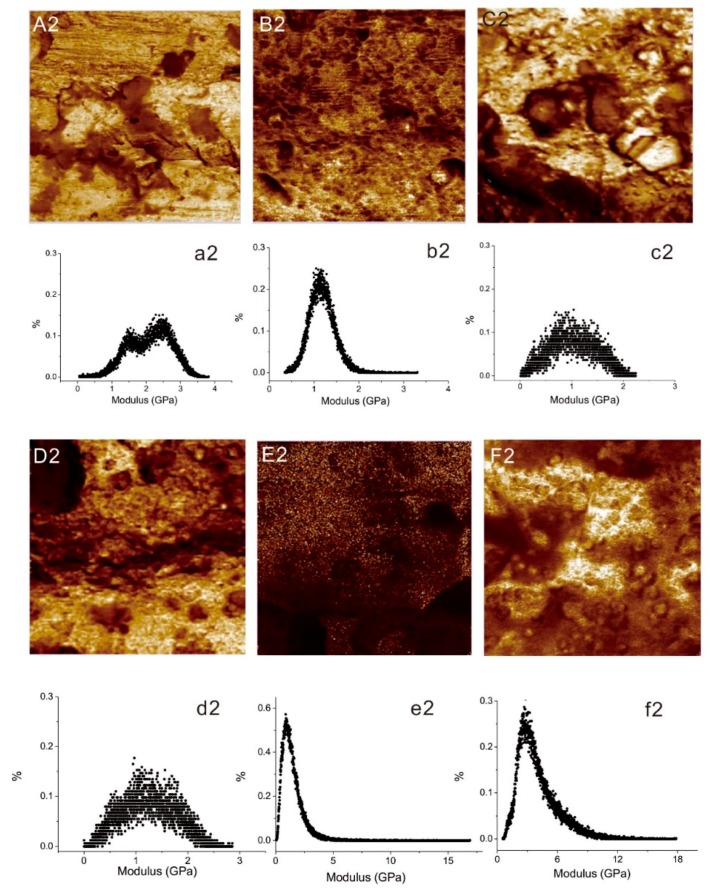
AFM Derjaguin–Muller–Toporov (DMT) modulus maps of a 3 × 3 μm^2^ region, and the modulus distribution graphs of (**A2**,**a2**) Al alloy; (**B2**,**b2**) MP; (**C2, c2**) MPAL; (**D2**,**d2**) MPALSi; (**E2**,**e2**) P2; (**F2**,**f2**) P2Si.

**Figure 6 materials-12-03363-f006:**
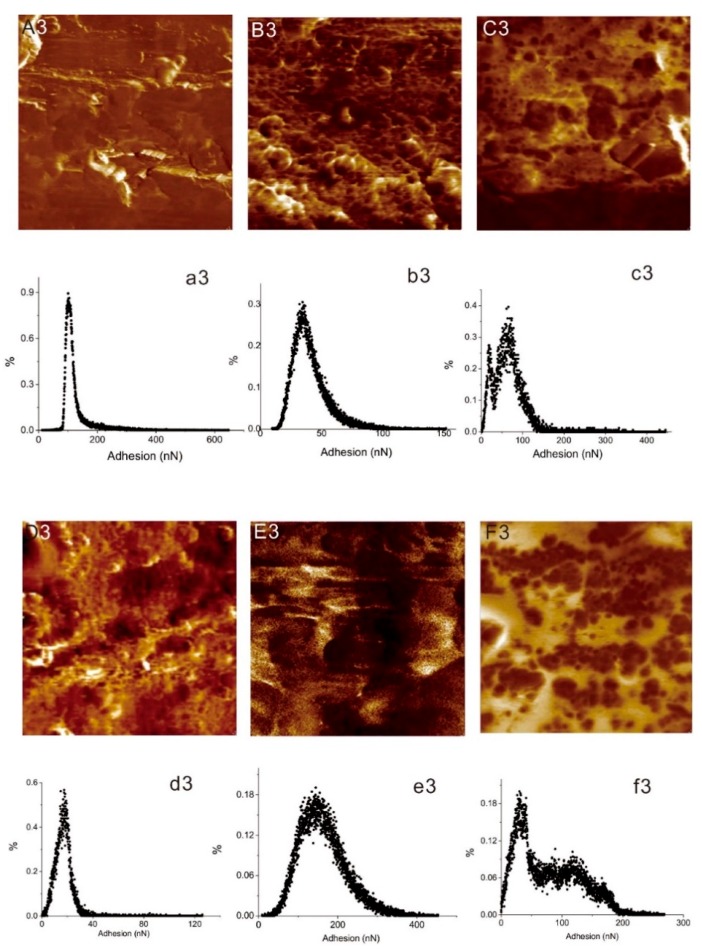
AFM adhesion maps of a 3 × 3 μm^2^ region, and the adhesion force distribution graphs of (**A3**,**a3**) untreated Al alloy; (**B3**,**b3**) MP; (**C3**,**c3**) MPAL; (**D3**,**d3**) MPALSi; (**E3**,**e3**) P2; (**F3**,**f3**) P2Si.

**Figure 7 materials-12-03363-f007:**
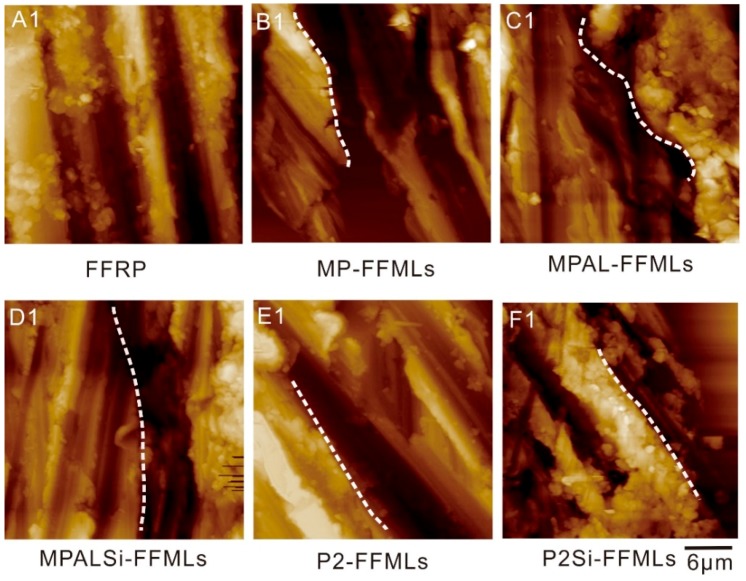
AFM height images of the flax fiber-reinforced plastic (FFRP) and the five type of modified flax fiber–metal laminates (FFMLs).

**Figure 8 materials-12-03363-f008:**
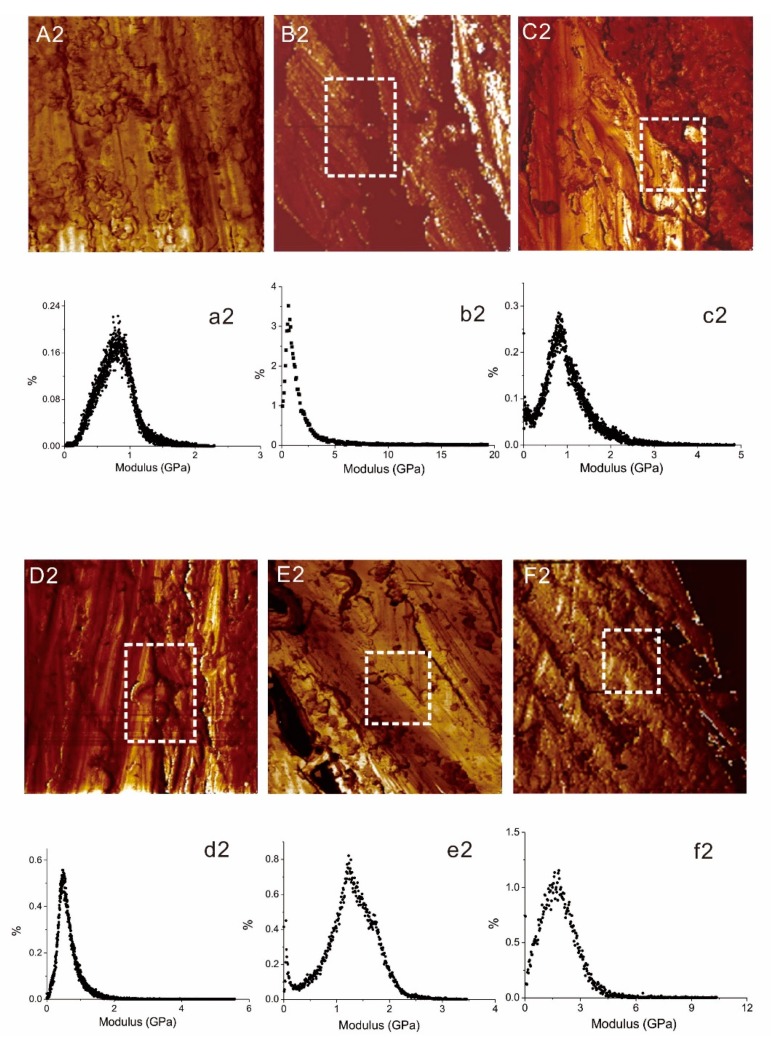
AFM DMT modulus maps of a 30 × 30 μm^2^ region, and the modulus distribution graphs of: FFRP (**A****2**,**a2**); MP-FFML (**B2**,**b2**); MPAL-FFML (**C2**,**c2**); MPALSi-FFML (**D2**,**d2**) MPALSi; P2-FFML (**E2**,**e2**); P2Si -FFML (**F2**,**f2**).

**Figure 9 materials-12-03363-f009:**
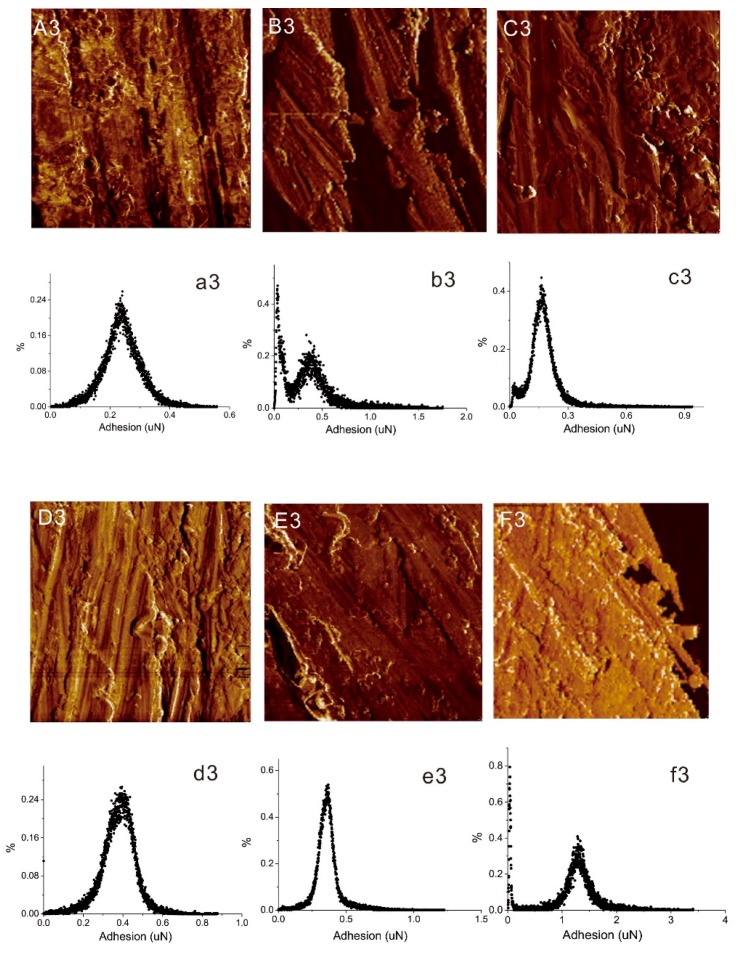
AFM adhesion force maps of a 30 × 30 μm^2^ region, and adhesion force distribution graphs of (**A3**,**a3**) FFRP; (**B3**,**b3**) MP-FFML; (**C3**,**c3**) MPAL-FFML; (**D3**,**d3**) MPALSi-FFML; (**E3**,**e3**) P2-FFML; (**F3**,**f3**) P2Si-FFML.

**Figure 10 materials-12-03363-f010:**
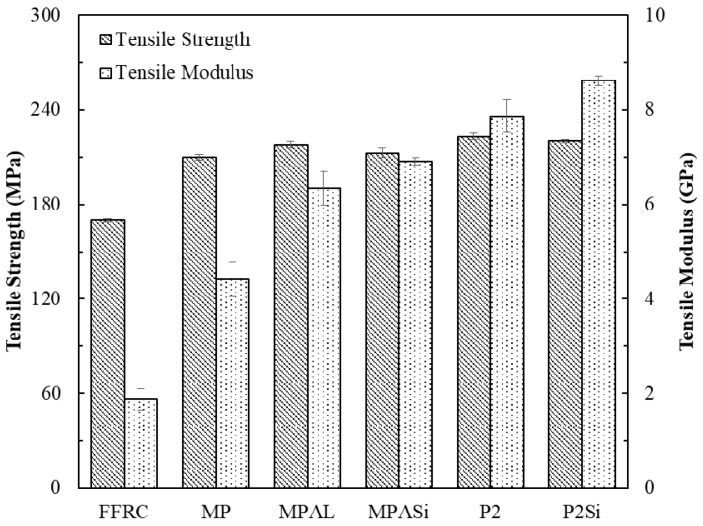
Tensile strength and modulus of FFRP and five types of modified FFMLs composites.

**Figure 11 materials-12-03363-f011:**
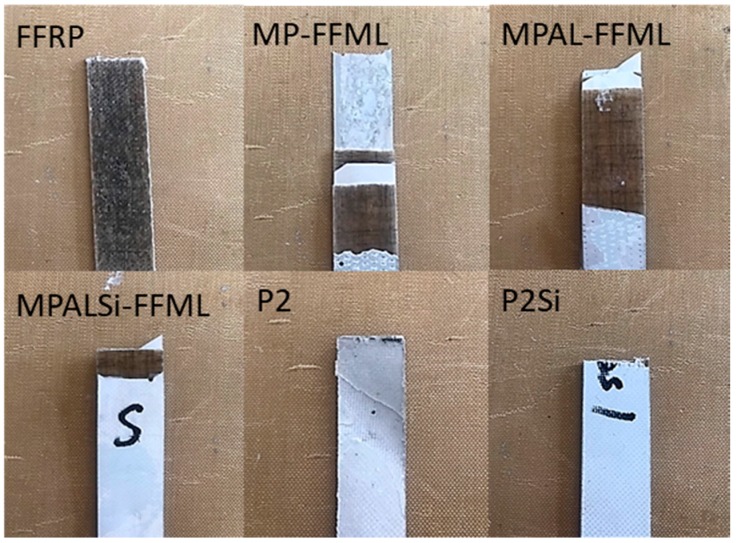
The damaged area of FFRP and five types of modified FFMLs following tensile tests.

**Figure 12 materials-12-03363-f012:**
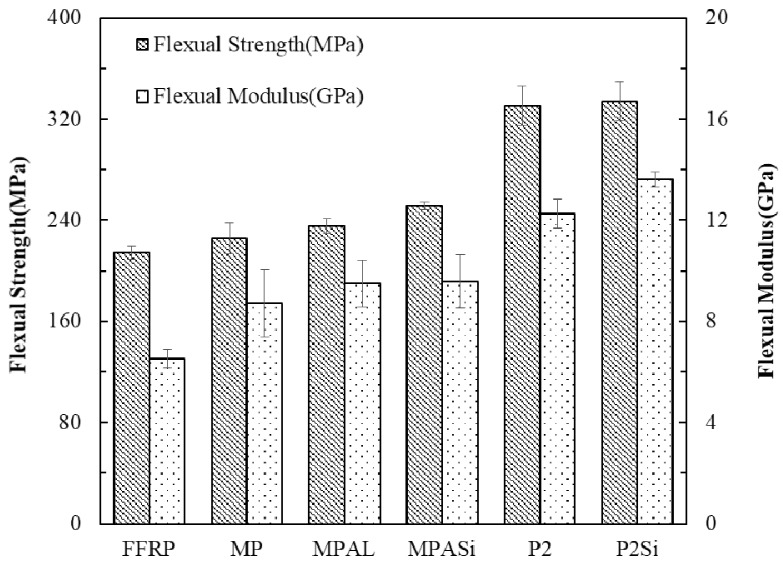
Flexural strength and modulus of FFRP and five types of surface-treated FFMLs composites.

**Figure 13 materials-12-03363-f013:**
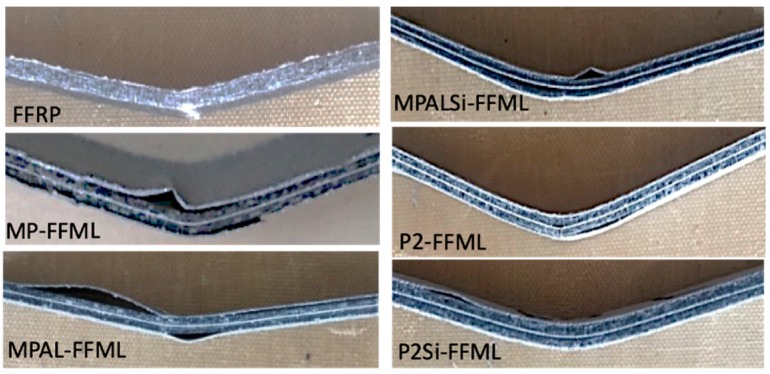
The side view of FFRP and five types of surface-treated FFMLs following three-point bending tests.

**Table 1 materials-12-03363-t001:** The different surface treatments applied to the Al alloy sheets.

Sample ID.	Surface Treatment Method
MP	Mechanical polishing (T1)
MPAL	Mechanical polishing and the alkali solution immersion (T1 + T2)
MPALSi	Mechanical polishing, alkali solution immersion, and silane coupling agent surface treatment (T1 + T2 + T4)
P2	P2 etching (T3)
P2Si	P2 etching and silane coupling agent surface treatment (T3 + T4)

**Table 2 materials-12-03363-t002:** The parameters of flax fiber–metal laminates (FFMLs) and flax fiber-reinforced plastic (FFRP) composites.

Laminates	Density (g/cm^3^)	Fiber Volume Percentage (%)	Metal Volume Percentage (%)	Thickness (mm)
FFML	1.42	22%	18%	2.45 ± 0.10
FFRP	1.20	36%	/	2.58 ± 0.23

**Table 3 materials-12-03363-t003:** The roughness data of the different surface modified Al alloy sheets obtained using an atomic force microscope (AFM).

Sample ID	Rq	Ra
Al alloy	72.9	46.7
MP	21.3	17.1
MPAL	82.5	61.5
MPALSi	58.4	42.3
P2	78.3	62.4
P2Si	46.8	36.3

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
