# Peer review of "Evaluation of Nano-Mechanical Behavior on Flax Fiber Metal Laminates Using an Atomic Force Microscope"

_materials, 2019, doi:10.3390/ma12203363_

Round 1

Reviewer 1 Report

The authors studied the influence of different treatment methods applied to Al alloys layers to improve their adhesion to plant fiber reinforced composites and to obtain hybrid structures with interesting performances.

Specifically, the effects induced by five different metal surface pre-treatments were evaluated in terms of surface morphological aspects and nano-mechanical performances determined by atomic force microscopy.

The research is interesting given potentials already documented for the so-called fiber metal laminates but the manuscript must be strongly revised before it can be considered acceptable.

In general, a careful re-reading of the manuscript is advised both to remove several typing errors still present in the text and to rewrite some sentences not readily understandable even for readers of the sector.

Some suggestions are explained below:

In the abstract row 21, for the first time P2 etching is written. In my opinion, for clarity of exposure to readers who are not familiar with these superficial pre-treatments it is advisable to specify, at least in correspondence with the first quotation, what is meant. Introduction, lines 71-75. The sentence is not clear. Please, rephrase. Materials and Methods. Materials, line 92. Please check the area density. Maybe it is 0.021 g /cm2. Surface morphology of materials, line 149. The improved interfacial adhesion absolutely cannot be identified with the formation of interpenetrated polymer networks which, as well established, indicate structures characterized by two or more networks interlaced on a polymeric scale. Can authors prove the occurrence of this situation? Please explain better what you mean. Paragraph 3.4. Lines 274-277. The sentence is not clear. Please, rephrase.

Overall, despite the adequacy of the applied characterization techniques, major revisions are strongly required to improve the quality of the paper.

Author Response

The authors would like to thank the precious advices from the reviewer. This paper has been carefully modified according to the reviewer’s advices.

In the abstract row 21, for the first time P2 etching is written. In my opinion, for clarity of exposure to readers who are not familiar with these superficial pre-treatments it is advisable to specify, at least in correspondence with the first quotation, what is meant.

Response: “the sulphuric acid-ferric sulphate-based treatment (P2 etching)” is added in the abstract.

Introduction, lines 71-75. The sentence is not clear. Please, rephrase.

Response: The sentence was revised as “With the development of atomic force microscopy, several researchers have applied AFM technique on accessing mechanical properties of glass fiber reinforced composite laminates at submicron or nano-scale [17-19].”

Materials and Methods. Materials, line 92. Please check the area density. Maybe it is 0.02 g /cm2.

Response: The areal density of the flax fabric is actually 200g/m2, and it was modified in the paper.

Surface morphology of materials, line 149. The improved interfacial adhesion absolutely cannot be identified with the formation of interpenetrated polymer networks which, as well established, indicate structures characterized by two or more networks interlaced on a polymeric scale. Can authors prove the occurrence of this situation? Please explain better what you mean.

Response: It is true that the formation of a silane film on flax fiber can not indicate the formation of interpenetrated polymer networks or the improvement on interfacial adhesion. Because this paper is mainly focused on the effect of metal surface treatments on the nano-mechanical performance of FFMLs, therefore, this paragraph was modified as below and quote some others study results on silane treated plant fibers:

“First of all, the silane treatment on flax fiber causes the formation of covalent bonds between the organic functional group of the silane and the hydroxyl of flax fiber. A silane film was found formed on the surface of the flax fiber as show in Figure 2. The interfacial adhesion between untreated flax fibers and hydrophobic epoxy are very weak due to the presence of hydroxyl and other polar groups in the flax fiber. The hydrophilic nature of the flax fibers can be minimized with silane coupling agent treatment, it can increase the wettability of the fibers with matrix and the interfacial bond strength. There have been many researches proved that the silane treated plant fiber and epoxy could form interpenetrating polymer networks with the characterization methods of XPS, AFM and so on, and the interface adhesion between flax fiber and resin within the composite were clearly enhanced [26,27].”

Paragraph 3.4. Lines 274-277. The sentence is not clear. Please, rephrase.

Response: This sentence was revised as “In addition, the tensile modulus of FFRP and five type of modified FFMLs show good agreement with the DMT modulus of these composites, which is the P2Si-FFMLs provide the highest modulus among these composites.”

Reviewer 2 Report

The study put forward by Zehua et al.,  presents several surface modifications on flax fiber and aluminum alloy sheets in order to improve the interlaminar properties of the FFMLs composites. The authors described various modification techniques and conducted material characteristics. In general the paper is of an average value with respect to broadening knowledge and extending methods of plant fiber reinforced composites. The way the results are presented is clear and legible. Figures and tables understandable. The description of the results forms a unified whole creating a logical cause and effect sequence. Stylistics of language at a satisfactory level.

I recommend the paper to be publish in the Materials journal after making minor revision.

Below are some comments with a request to explain and respond to them:

3 Line 95, Please add more specific description of composites preparation 3 Line 117, Please provide the characteristics of the silane used for modification 4 Line 124, Please add more specific description of laminates fabrication 5 Line 144, Please add the name and manufacturer of the testing machine

Author Response

The authors would like to thank the precious advices from the reviewer. This paper has been carefully modified according to the reviewer’s advices.

 Line 117, Please provide the characteristics of the silane used for modification

Response: The silane used in this study is 3-Phenyl-aminopropyl-trimethoxy-sliane, and it was added in the paper.

Line 124, Please add more specific description of laminates fabrication

Response: More details of laminate fabrication was given in the paper “The layups of flax fabrics and thin Al alloy sheets were placed on the top of a smooth glass table between two sheets of peel plies. The resin system was degassed in a vacuum oven for 30 min to remove air bubbles before perfusion. Once the vacuum was surely set, the resin was impregnated in the mold and the flow was evenly distributed across the plate. The laminates were then cured at room temperature for 24 h and post-cured at 120 °C for 2 h.”

Line 144, Please add the name and manufacturer of the testing machine

Response: Tensile and flexural tests were performed on both of FFMLs and FFRP samples using a universal testing machine (CSS-20N, Shenzhen Wance Co.Ltd., China) according to ASTM D638 and ASTM D790, respectively.

Round 2

Reviewer 1 Report

The authors, according to my suggestions, have reviewed the manuscript and the new version can be considered acceptable.